# Pixel Representation Augmented through Cross-Attention for High-Resolution Remote Sensing Imagery Segmentation

Yiyun Luo [1,2], Jinnian Wang [1,2,*], Xiankun Yang [1,2], Zhenyu Yu [1,2] and Zixuan Tan [1,2]

1 School of Geography and Remote Sensing, Guangzhou University, Guangzhou 510006, China
2 Centre for Remote Sensing Big Data Intelligence Applications, Guangzhou University, Guangzhou 510006, China
* Correspondence: jnwang@gzhu.edu.cn

**Abstract:** Natural imagery segmentation has been transferred to land cover classification in remote sensing imagery with excellent performance. However, two key issues have been overlooked in the transfer process: (1) some objects were easily overwhelmed by the complex backgrounds; (2) interclass information for indistinguishable classes was not fully utilized. The attention mechanism in the transformer is capable of modeling long-range dependencies on each sample for per-pixel context extraction. Notably, per-pixel context from the attention mechanism can aggregate category information. Therefore, we proposed a semantic segmentation method based on pixel representation augmentation. In our method, a simplified feature pyramid was designed to decode the hierarchical pixel features from the backbone, and then decode the category representations into learnable category object embedding queries by cross-attention in the transformer decoder. Finally, pixel representation is augmented by an additional cross-attention in the transformer encoder under the supervision of auxiliary segmentation heads. The results of extensive experiments on the aerial image dataset Potsdam and satellite image dataset Gaofen Image Dataset with 15 categories (GID-15) demonstrate that the cross-attention is effective, and our method achieved the mean intersection over union (mIoU) of 86.2% and 62.5% on the Potsdam test set and GID-15 validation set, respectively. Additionally, we achieved an inference speed of 76 frames per second (FPS) on the Potsdam test dataset, higher than all the state-of-the-art models we tested on the same device.

**Keywords:** land cover classification; transformer; cross-attention; object embedding queries

## 1. Introduction

Natural and social properties of the land surface can be presented by land cover. The land cover map shows the local or overall landscape state of a region, based on which we are able to study environmental change trends. On the other hand, it can be used to assess urban development and estimate the extent of the impact of natural disasters. With the information from land cover maps, city managers can make better decisions and anticipate the impact of those decisions on the region in advance [1].The identification of land types to obtain high-precision land use maps has been a longstanding research topic. The pixel-level classification for remote sensing images has been widely used over the past decades [2]. It relies on the spectral features for classification, such as vegetation index, building index, and others. Then, the inversion of land use type information was obtained by the analysis of multispectral satellite images [3–5]. Another classification method is to build a classifier to explore the semantic features between pixels [2,3,6]. Currently, with the advances in sensors and computer technology, the remote sensing images have higher spatial resolution and more complex pixel representation. For such images, the traditional image interpretation methods [7,8] may not meet the efficiency and accuracy of the real application. As a result, researchers in the remote sensing community have shifted their attention to methods-based deep learning. With robust feature modeling capabilities, deep

learning has achieved unprecedented progress in many tasks in remote sensing, such as land cover classification, scene classification, and object detection [9].

The trend toward deep learning began in 2012 when AlexNet [10] won the ImageNet Large Scale Visual Recognition Challenge, marking the shift of image tasks from the era dominated by machine learning methods to that of deep convolutional neural networks (DCNNs). Limited by the classification task, AlexNet could only output a probability of the sample as part of a category. It could not judge the category of each pixel within the whole image. The FCN [11] model proposed a pixel-level semantic segmentation by encoding-decoding convolutional neural networks and unveiled the key problems in semantic segmentation: loss of spatial information, insufficient perceptual field, and missing contextual information [12]. UNet [13] combined high-level semantic and fine-grained information and the convolutional deepening idea, and effectively improved the performance of the semantic segmentation network for large-sized graphics. The proposed PSPNet [14] is a PPM module to obtain contextual information in different regions. SegNet [15] achieved a significant increase in inference speed by replacing convolutional layers with empty convolution [14]. DeepLab [16], obtaining large perceptual fields through dilated convolution and the large kernel in its ASPP module, was also used for context information capture. HRNet can keep spatial information during the pixel decoding by keeping high-resolution representations through the whole pipeline. All these networks are based on the FCN's convolutional neural network codec architecture.

Due to the limitations of convolutional kernels, they can accept contextual information over only a short range, as shown in Figure 1c. As a result, previous studies have reported various types of attention mechanisms to acquire better long-range information [17], although they may cost more inference time. PSANet [18] developed a point-by-point spatial attention module for the dynamic capture of global context information. DANet embedded both spatial attention and channel attention. VIT [19] applied the transformer [20] architecture based solely on attention mechanisms to image information extraction and pioneered a visual recognition method based on image patches. A variety of visual transformer-based approaches have subsequently emerged, such as the Swin Transformer [21] and SETR [22]. And SegFormer [23] specifically designed a simple decoder for the transformer backbone, using only six linear layers for per pixel prediction, effectively exploiting the long-range semantics of the transformer encoder. The transformer and attention mechanism effectively improve the performance of various downstream tasks, including semantic segmentation, but the tradeoff is the large dataset and a significant amount of training time.

For remote sensing imagery segmentation, high background complexity and indistinguishable classes are two challenging problems. As shown in the first row of Figure 1, the white boxes show multiple objects in the background, while the yellow boxes show the similarity between low vegetation and trees. Feature pyramid or image patch-based approaches cannot further improve performance because they focus only on pixel representations and lack deeper supervision of category objects. PFNet [24] adopts the point-wise affinity propagation module in the pixel decoder, which performs sparse mapping during forward between, improving training efficiency while reducing a large amount of noise from the background category. However, it does not consider interclass feature distinctions, and this results in ineffective recognition of indistinguishable classes. To enhance the representation of each pixel, MaskFormer [25] proposes a new idea for the panoramic segmentation task inspired by encoding category object features; it encodes object features into learnable queries, and then optimizes the binary mask and determines its class by object queries. It should be highlighted that the encoding of the image category representations is much less computationally intensive than pixel-by-pixel attention.

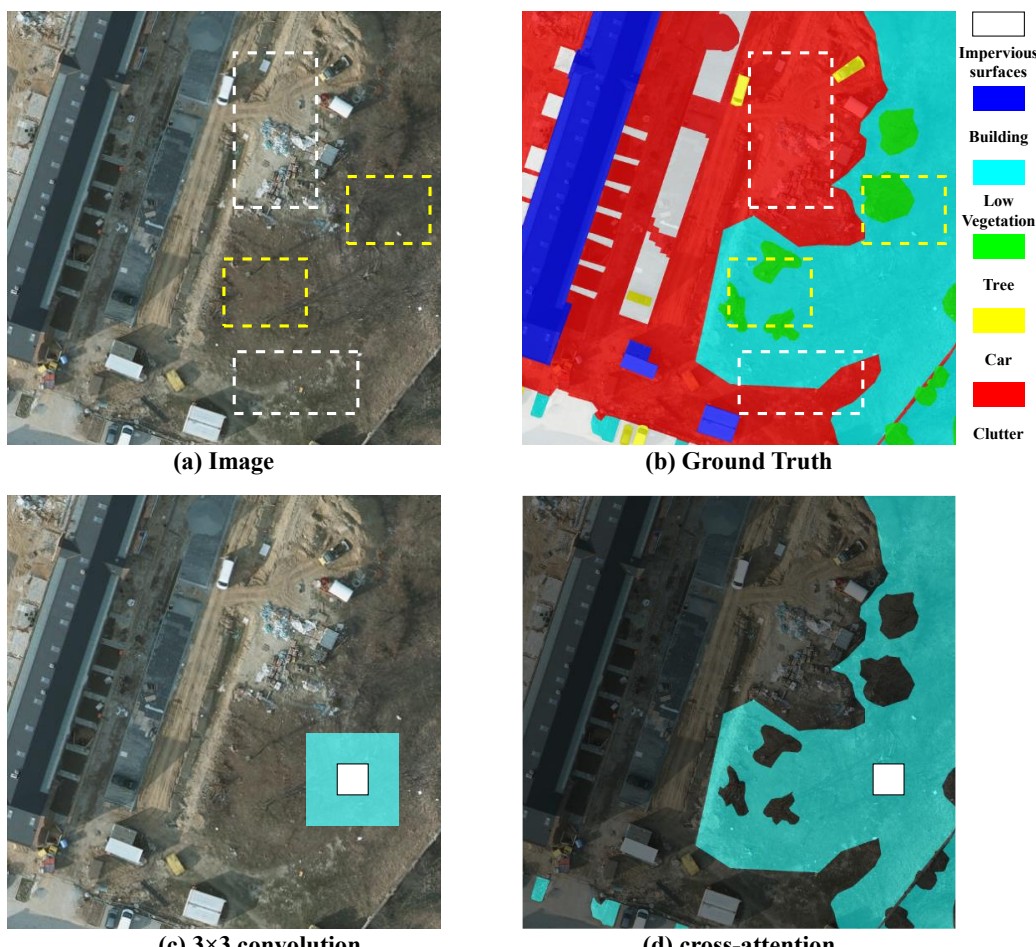

**Figure 1.** Illustration of a remote sensing image sample and cross-attention. (**a**) Image from the Potsdam dataset, (**b**) ground truth of the image. The sample pair presents the complex background: there are many objects with different features in the background category, and interclass similarity: overlap of the same features between different classes. The second row represents the comparison in the context extraction method between 3 × 3 convolution and cross-attention: (**c**) nine pixels near the sample point; (**d**) a set of pixels from the same class.

Based on the aforementioned excellent works, we proposed a cross-attention-based pixel augmented model to handle the problems stated above. It aims to maximize the use of category spatial information in the image, as shown in Figure 1d. Our main contributions are as follows:

1.  We propose an object-embedding queries module for augmenting the category representation of each pixel, which can model complex category features, and effectively improve the segmentation accuracy of the model for indistinguishable classes and background. The effectiveness of this module is verified in two remote sensing segmentation datasets with different resolutions, the Potsdam dataset and GID-15 dataset.

2.  Employing the cross-attention and transformer module to extract long-range semantic features in images. Both category representation decoding and encoding used cross-attention, but the details of their settings were a little different (See Sections 3.2 and 3.3).

3.  We designed a simplified feature pyramid for fusing the hierarchical features that were extracted by the backbone network at different stages; it reduces a large number of model parameters and provides high-quality representation for cross-attention during category representation decoding.

## 2. Related Works

Our method focuses on several key points: semantic segmentation on high-resolution remote sensing images; pixel decoding; object embedding queries; transformer module; interaction of cross-attention semantic features with spatial context information; and booster training with auxiliary output.

### 2.1. Semantic Segmentation

Early works on land cover classification primarily used surface features such as the color and texture of images for manual visual interpretation classification [26,27], or simple supervised classification [28,29] and unsupervised classification [30]. After neural networks became popular, a series of network models applied to remote sensing images were developed [6]. To date, the most widely used method for image characteristic extraction is still deep network classification based on image spatial information and contextual features. This technique is often referred to as deep learning-based semantic segmentation, whose pipeline is based on the FCN's encoder-decoder structure. FCN replaced fully connected layers with convolutional layers and introduced deconvolution, the improvement of more than 20% compared with previous methods on the PASCAL VOC dataset, and solved the problem of CNNs model converging slowly and demanding fixed-size input. There are numerous FCN-based models for land cover classification, building detection, agricultural monitoring, and other scenarios. However, the accuracy of these methods is still affected by the problem of intra-class diversity and inter-class similarity Recently, some works attempted to obtain semantic information from different categories by extracting additional pixel-wise information. HPSNet [31], for example, utilized a mini-branch in the part of the encoder for global feature optimization, and demonstrated on the GID-5 and GID-15 datasets that the use of mini-branches in the network optimization process resulted in better pixel-level optimal path selection for coding the distribution of land cover.

The method we proposed introduces cross-attention to the decoder, which effectively embeds the category representation into each pixel of the samples, and implements the segmentation of indistinguishable classes.

### 2.2. Pixel Decoder

Our pixel decoder contains two parts: feature fusion and feature upsampling. The classical means of feature fusion are addition and concatenation [11,32]. Feature pyramid networks (FPNs) [33] fuse low-resolution features with spatial information and high-resolution features with texture features by addition, alleviating the problem of multi-scale target missing. Additionally, the feature alignment module proposed by FaPN [34] added an additional bias to the fusion of features at different resolutions to match their positions, which reduces the error generated by features during upsampling. HRNet [35] concatenates features of different resolutions obtained after multi-resolution fusion convolution, retains rich semantic information and more precise spatial information., and has achieved excellent scores on a large number of tasks, including semantic segmentation, human pose estimation, and object detection.

To obtain faster speed while reflecting the effectiveness of our designed object embedding queries and cross-attention, instead of using a complex pixel decoder with a large number of parameters, we simply fuse multiresolution features by concatenation from top to bottom of the feature pyramid.

### 2.3. Contextual Information

Unlike remote sensing scene classification [36], the semantic segmentation task concerns the spatial details of the image and contextual information, and the problem often faced in remote sensing images lies in the large differences in target scales and uneven distribution of categories. To solve this kind of problem of spatial context information extraction, BiSenet [37] set up two branching structures to extract detailed information and

semantic information and connected them. This enhanced their feature representations through a bidirectional aggregation layer. DANet [38] sets up a channel attention branching network to capture correlations between spatial locations. ACFNet [39] proposed a class feature representation of Class Center, which first obtained coarse segmentation results and then implemented feature-to-class mapping at each pixel point for final refinement output through a class attention mechanism, where the formation of class region features was unsupervised. Based on this category-level contextual feature embedding, HRNet [35] fused the high-resolution image features and low-resolution image features stored in its backbone to generate category object representations and regional pixel representations, and supervise the generation of coarse segmentation maps by ground truth, solving the problem that coarse segmentation accuracy could not be improved.

We designed flexible object embedding queries that can flexibly transform the dimension according to the characteristics of the target remote sensing image to overcome this challenge.

### 2.4. Transformer and Cross-Attention

The original transformer model was proposed for sequence-to-sequence machine translation tasks. It consisted of multiple encoder-decoder architectures. There are several different application approaches in computer vision. VIT [19] used only encoders; Mask-Former [25] used only decoders; and DETR [40] used both encoders and decoders. The transformer model consisted of multiple encoder-decoder architectures where the encoder is divided into two parts: self-attention and feed-forward networks. The decoder adds a cross-attention layer between these two parts compared with the encoder, which is used to aggregate the encoder's output and the input features of the decoder [20].

Global contextual information is significant, and attention can effectively improve the performance of long-range semantic modeling. CCNet [41] extracted the correlation tensor from the underlying features to obtain the correlation between a pixel and its horizontal and vertical pixel points, and combined it with vector-containing semantic features. OCRNet [35] extracts the relationship between each pixel and the object region in which the pixel is located, representing object features as category queries, and assigns the object features to each pixel in the sample using a transformer decoder-like structure with cross-attention. OCRNet provides an efficient method for contextual feature aggregation.

All these methods use cross-attention to generate per-pixel features at different scales or different categories, and output predictions with generation of the high-resolution feature map. Obviously, the pixel-by-pixel category features encoding leads to a spike in the number of parameters when the model restores the resolution. In our method, we solved this problem by decoding and encoding category representations by two different cross-attentions, respectively.

### 2.5. Auxiliary Loss

Auxiliary tasks are commonly used a priori in multitask learning (MTL) [42] to facilitate the process of learning optimization. In semantic segmentation tasks, like BiSenet [37] and PSPNet [14], auxiliary loss functions are used in convolutional layers to make use of the idea of hierarchical feature progression in the semantic segmentation encoder-decoder to make predictions on different layers of features so that the features can be effectively mined and the weights of primary and auxiliary losses balanced with the help of parameters. In contrast, the network DETR [40] and MaskFormer [25] that use the transformer architecture give the loss ensemble by a redundant class token, and the token with the smallest loss is selected to participate in the loss balancing, preventing the problem of uneven category distribution in image patches. SETR [22] upsampled the features to the original resolution output as an auxiliary loss. In summary, the auxiliary loss can optimize the network training process, and has the advantage of not increasing the time spent in the inference phase.

## 3. Methods

The remote sensing imagery-based land cover classification problem is solved by obtaining the probability distribution $y$ of $H \times W$ pixels in an image and assigning it to one of the species in one of the label categories $K_i$ with the highest probability. Formally,

$$y = \left\{ p_i \mid p_i \in \Delta^K \right\}_{i=1}^{HW}, \tag{1}$$

where $\Delta^K$ denotes a $K$-dimensional probability simplex. $p_i$ denotes the probability that pixel $i$ belongs to a category, and $y$ is the set of $p_i$ output by the segmentation model of all pixels in the sample image. To train a segmentation model for pixel-by-pixel classification, each pixel is assigned to a ground truth category $y^{gt}$. As in Equation (2):

$$y^{gt} = \left\{ y_i^{gt} \mid y_i^{gt} \in 1, \dots, K \right\}_{i=1}^{HW}, \tag{2}$$

where $y^{gt}$ indicates the category to which each pixel in the sample image is assigned. Then, the loss is calculated by the loss function. Based on the pixel category cross-entropy as Equation (3):

$$\mathcal{L}_{pixel-cls}\left(y, y^{gt}\right) = \sum_{i=1}^{HW} -\log p_i\left(y_i^{gt}\right) \tag{3}$$

Accurate and efficient acquisition of pixel class probability $p_i$ is the main task of segmentation. An overview of our method to obtain segmented prediction is shown in Figure 2, and our proposed scheme is as follows:

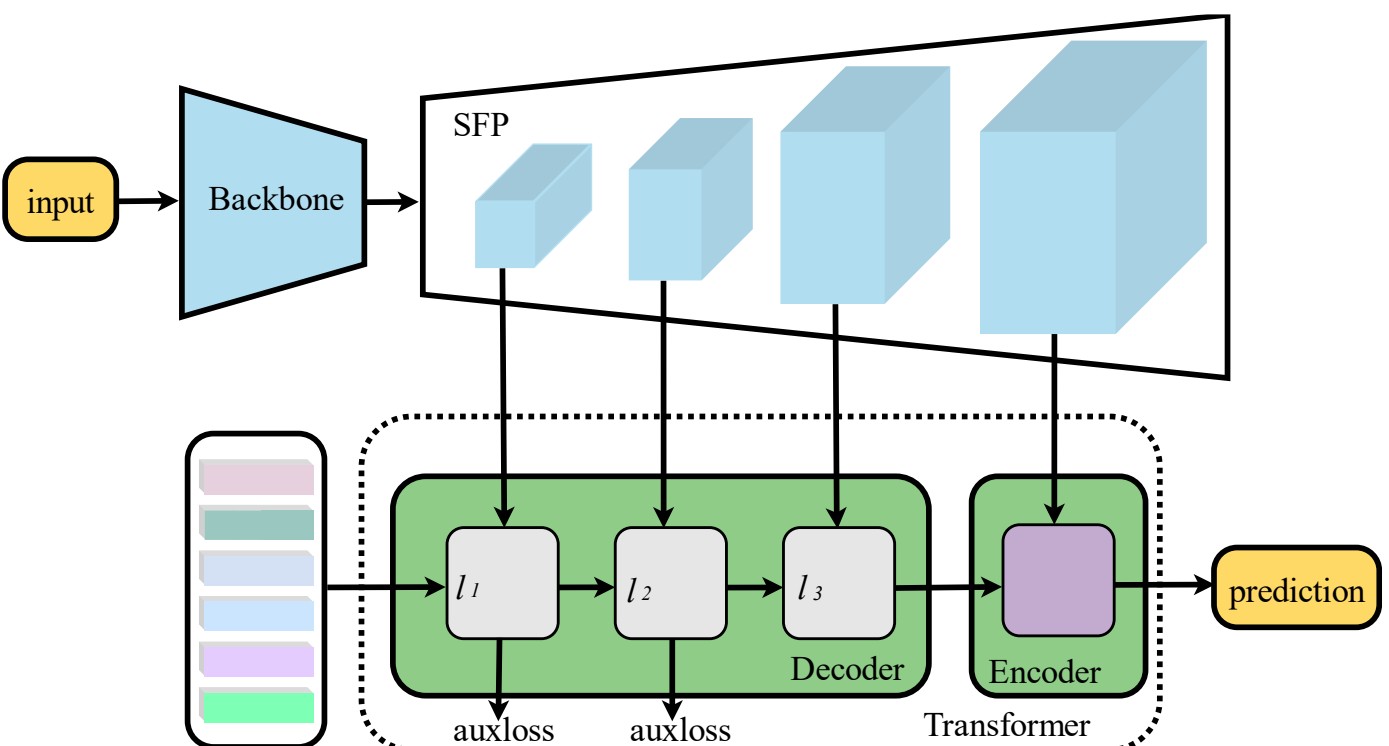

**Figure 2.** Neural Network overview. We propose an innovative transformer module with cross-attention that utilizes multiscale features from a pixel decoder (Section 3.3). Additionally, we proposed object embedding queries to aggregate class features (Section 3.2). Stage 4 in the SFP and transformer encoder layers for auxiliary loss are omitted in this figure for readability.

(1) Feature pyramid-based pixel decoding; (2) generating learnable object embedding queries $D_c$ and extracting category-contextual features by decode cross-attention;

(3) augmenting the per-pixel feature representation by aggregating category-contextual information with encoder cross-attention.

### 3.1. Pixel Decoder

The backbone used for semantic segmentation usually constructs a collection of features from high to low resolution. Based on the idea of the feature pyramid [33], we fused features of different resolutions by one feature space transformation (by a $1 \times 1$ convolutional layer) and one semantic feature extraction (by a $3 \times 3$ convolutional layer) and upsample them to the higher-level features to achieve the fusion of multiscale features. The pixel decoder module can be written as Equation (4):

$$x_i = \begin{cases} \text{Conv}_{3\times3}(\text{Conv}_{1\times1}(X_4)), & i = 1 \\ \text{Conv}_{3\times3}(\text{Conv}_{1\times1}(X_{5-i}) + \text{Upsample}(x_{i-1})), & 2 \leq i \leq 4 \\ \text{Conv}_{1\times1}(\text{Concat}(x_1, x_2, x_3, x_4)), & i = 5 \end{cases} \quad (4)$$

We call this module the simplified feature pyramid (SFP), and the structure is shown in Figure 3, which serves to aggregate the hierarchical features extracted by the backbone. When the image size of the input model is $H \times W$, the backbone extracts a series of features with different resolutions $X_i \in R^{C_X \times \frac{H}{S} \times \frac{W}{S}}$, where $C_X$ is the number of feature channels and $S$ is the downsampling multiplicity. The maximum downsampling rate chosen in our method is $S = 32$, and $x_i$ are the features after SFP. $\text{Conv}_{1\times1}$ is the feature space transformation function implemented by $1 \times 1$ convolution. $\text{Conv}_{3\times3}$ is the feature extraction function implemented by $3 \times 3$ convolution $\rightarrow$ GN $\rightarrow$ ReLU. Concat is the function that concatenates each feature in the channel dimension. Before concatenation, we sample all resolution features down to $\frac{H}{4} \times \frac{W}{4}$.

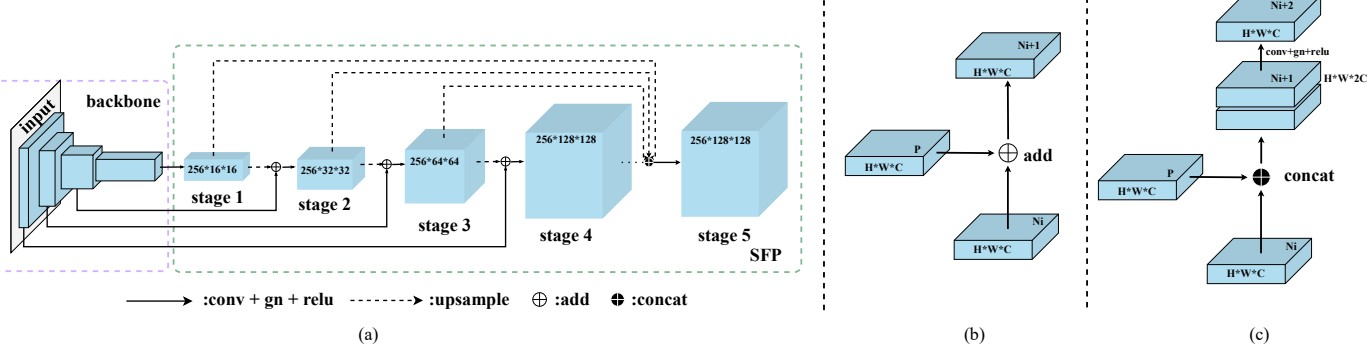

**Figure 3.** Hierarchical structure for spatial feature extraction. (**a**) The simplified feature pyramid (SFP) module pipeline. (**b**) The add method in SFP. (**c**) The concatenation method in SFP.

### 3.2. Object Embedding Queries and Decoder Cross-Attention

The label that each pixel has expressed is the semantic label of the object in which the pixel is located. Based on this concept, we use a set of learnable object embedding queries (OEQs) *A* to aggregate the contexture features of all pixels through category objects.

As shown in Figure 4 (left), the object embedding queries structure the image into *N* object regions, represented as *A*, train it interactively with the four stages of features expressed through the feature fusion module SFP in the transformer decoder module with different stages of pixel-level features for the purpose of aggregating each pixel representation in the image.

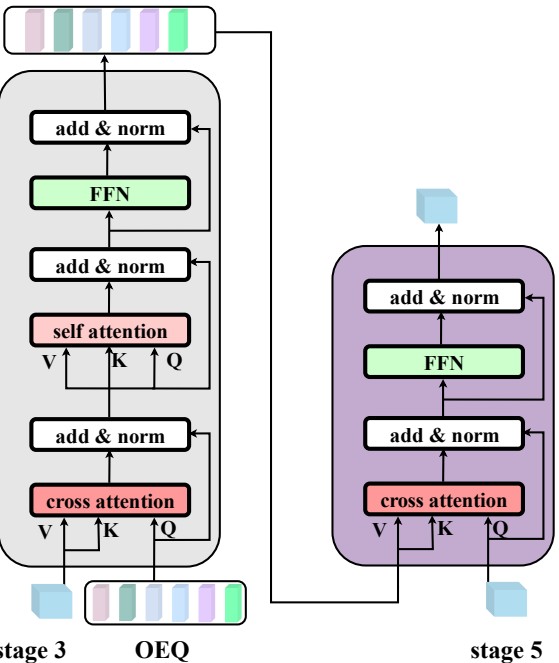

**Figure 4.** Illustration of the transformer decoder (**left**) and the transformer encoder (**right**) after SFP stage 3.

The transformer containing the decoder cross-attention used in this paper uses a similar setup as in DETR [40], with the $N$-dimensional object embedding queries initialized to a zero vector, a learnable position encoding attached to each query of the input, and an auxiliary loss function after each transformer. Additionally, considering the model inference time, we reduce the transformer stack and use only three transformer layers. In addition, we move the self-attention module to the back of the cross-attention module to avoid useless training at the beginning.

In the transformer decoder, long-range semantic modeling is performed by attention mechanisms. The attention mechanism usually means that the weight matrix containing similarity information is normalized by softmax and then applied to the transformed features by the dot product, and an output containing contextual information is obtained. In addition, we construct the weight mask, which is used to reduce the computational effort of the network and accelerate the convergence. Our object embedding queries are updated in the decoder cross-attention (with a shortcut layer) as Equation (5):

$$A_l = \mathcal{M}_l \left[ \sum_{j=1}^{C_{kv}} \alpha_{1,j}, \ldots, \sum_{j=1}^{C_{kv}} \alpha_{Nj} \right] V_j + A_{l-1} \tag{5}$$

where $l$ is the decoder layer index corresponding to the number of stages in SFP, $A_l \in R^{N \times C}$ represents the object embedding queries output by the $l$th layer decoder, and $\alpha$ is the attention *weight*. In Figure 5, this is computed as a softmax normalization of the dot-product between queries $Q_n$ and keys $K_j$ for querying the correlation between features and categories for each pixel (We set $N$ to a number greater than or equal to the number of classes, and $j$ is a feature embedding index). $\mathcal{M}_l$ is the binarized output of the previous transformer encoder layer by sigmoid activation (thresholded at 0.5). Moreover, attention weight $\alpha$ at the cross attention is as follows:

$$\alpha_{nj} = \frac{e^{\frac{1}{\sqrt{d}} Q_n^\top K_j}}{\sum_{j=1}^{C_{kv}} e^{\frac{1}{\sqrt{d}} Q_n^\top K_j}} \tag{6}$$

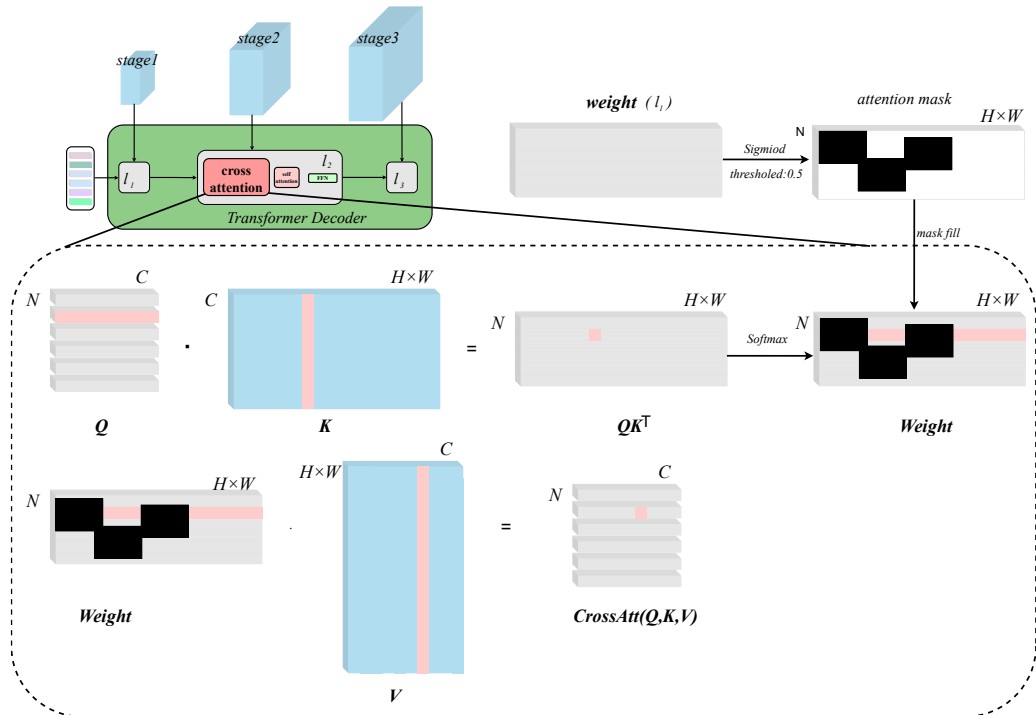

**Figure 5.** Illustration of the decoder cross-attention with an attention mask.

Here, $Q = f_Q(A_{l-1}) \in \mathbb{R}^{N \times C}$ and $A_0$ denote input object embedding queries to the transformer decoder. $K = f_K(\mathrm{x}i) \in \mathbb{R}^{H_l W_l \times C}$; $V = f_V(\mathrm{x}i) \in \mathbb{R}^{H_l W_l \times C}$, $f_K(\cdot)$ and $f_V(\cdot)$ are the linear transformations for mapping image features to different feature spaces, and $H_l$ and $W_l$ correspond to the resolution of the spatial feature $\mathrm{x}_i$ output from the pixel decoder module.

### 3.3. Encoder Cross-Attention

The encoder cross-attention is also contained in the transformer module (See Figure 4 (right)). Unlike the decoder cross-attention, it did not use the self-attention. Encoder cross-attention was used to augment each pixel representation output from the SFP by $N$ object embedding queries as in Figure 6 and Equation (7):

$$y_i = \rho\left(\sum_{n=1}^{N} w_{in}\delta(A_n)\right) \tag{7}$$

where $A_n$ is the output from the decoders, and $\rho(\cdot)$ and $\delta(\cdot)$ are two linear transformations with a ReLU activation. The attention weights $w_{in}$ between pixel $i$ and category $n$ are calculated as in Equation (8), which is similar to Equation (6):

$$w_{in} = \frac{e^{\nu(X_{5i}, A_n)}}{\sum_{n=1}^{N} e^{\nu(X_{5i}, A_n)}} \tag{8}$$

where $\nu(x, A) = \phi(x)^{\top}\psi(A)$ is the unnormalized relation function, and $\phi(x)$ and $\psi(A)$ are two linear transformations. Note that we used 3 decoder layers and added an encoder with the same settings after each decoder layer for the auxiliary output. In our experiments, only the last layer of the decoder output was very competitive.

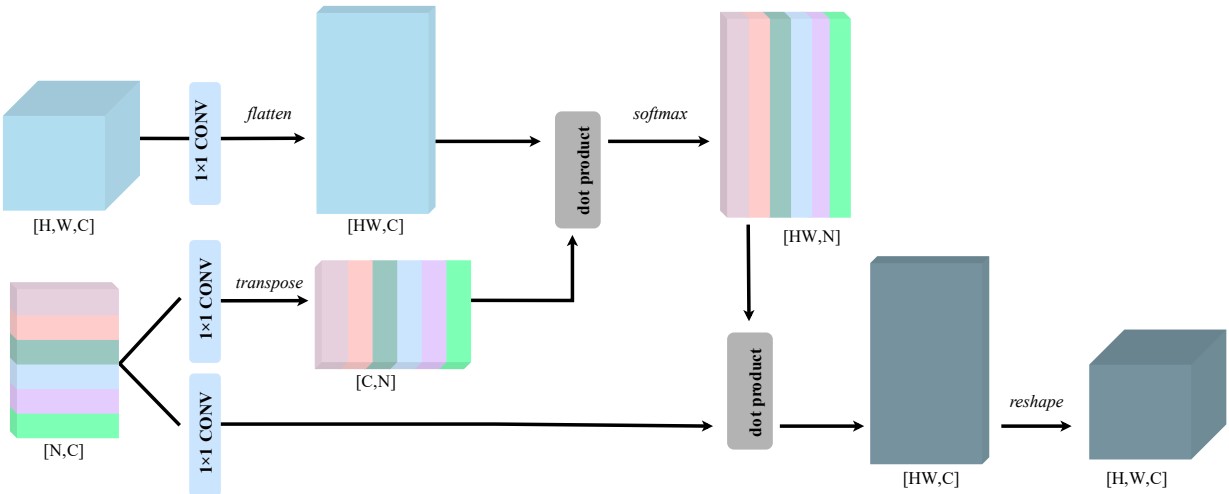

**Figure 6.** Illustration of the encoder cross-attention.

### 3.4. Loss Function

The average of focal loss [43] and dice loss [44] have been implemented in our work because of their potential to tackle class imbalance in remote sensing images. For example, the number of pixels on impervious surfaces and buildings is much larger than that on cars. These two loss functions have been computed under a binary classification after the one-hot encoder. $p_{ci}$ is the probability that the pixel belongs to class $c$, with the background of class $c$ probability being $1 - p_{ci}$. Moreover, the same loss functions are used in the auxiliary loss.

**Focal loss (FL)** focuses training on hard negatives and can be expressed as:

$$FL = -\frac{1}{NC} \sum_{i=1}^{N} \sum_{c=1}^{C} \left[ \alpha_c (1 - p_{ci})^{\gamma} ln(p_{ci}) \right] \tag{9}$$

where $\alpha$ and $\gamma$ are the modulating factors, $\alpha$ is used to increase the weight of the category with fewer samples in the loss function, and $\gamma$ is used to increase the weight of samples with larger classification loss in the loss function.

**Dice loss (DL)** is based on the Dice coefficient, which is used to gauge the similarity of two samples as follows:

$$DL = \sum_{c=1}^{C} \left[ 1 - \frac{\sum_{i=1}^{n} p_{ci} r_{ci}}{\sum_{i=1}^{n} p_{ci} + r_{ci}} - \frac{\sum_{i=1}^{n} (1 - p_{ci})(1 - r_{ci})}{\sum_{i=1}^{n} 2 - p_{ci} - r_{ci}} \right] \# \tag{10}$$

Here, $r_{ci}$ are the voxel values from the reference foreground segmentation. Furthermore, the second term on the right side of the above equation represents the predictive performance of the category foreground, and the third term represents the predictive performance of the background.

To optimize the learning process, we supervise the training of our proposed method using the auxiliary loss function, whose number depends on the number of layers of the transformer decoder L. The output of the whole network is calculated from the object query output of the last decoder layer. The weight between the auxiliary loss and main loss is balanced by the parameter $\alpha$ and the depth of the decoder. Each loss function is composed of the two loss functions mentioned above, and the loss ratio of the two loss functions results is 1:1. The final loss function can be written as Equation (11):

$$Loss(A;Y) = l_p(A_L;Y) + \alpha \sum_{i=1}^{L-1} i l_i(A_i;Y) \tag{11}$$

where $l_p$ is the main loss, $l_i$ is the auxiliary loss of the $i$th layer of the decoder, $A_i$ is the output of the transformer decoder, $L$ is the number of layers of the transformer decoder,

and *Loss* is the joint loss function. In the network, we only perform the inference of the auxiliary function in the training phase.

## 4. Experimental Results

In this section, we present the details of the datasets used and implementation of the model. Then, we performed an analysis of the performance to assess the impact of each component of the model through ablation experiments and a comparison with state-of-the-art models of the same type. The results reported here concern the final accuracy and inference speed of the model in different benchmark tests.

### 4.1. Datasets and Training Setting

The datasets used in this study are the ISPRS open source Potsdam [45] and Wuhan University open source GID-15 [46] remote sensing satellite dataset. Our model achieved state-of-the-art results on both datasets.

**Potsdam** provides 38 GRB three-channel images with 6000*6000 pixels at 5 cm resolution with a segmentation category of 5 classes, excluding No. 7_10, which showed labeling errors. We removed the wrong label and selected 20 of them as the training set, 8 as the validation set, and 9 as the test set. Each image was cropped to 512*512 small images with 5% overlap ratio for each small image when cropping.

**GID-15** provides pixel-level labels for 15 categories with reference to the Chinese land use classification standard. This dataset contains 10 sheets with 7200*6800 pixels within the dataset, covering a land area of 506 km$^2$. Because the distribution of land cover in each image is not similar, instead of dividing the dataset into a training set, validation set and test set, we selected the same type of crop as the Potsdam dataset, and then 50% of the selected dataset was used as the training set, 20% as the validation set, and the remaining 30% as the test set.

We trained our model using the PyTorch framework with an RTX3080 GPU with an image input size of 512*512 pixels. We adopted Visual Attention Net [46] as the backbone. Moreover, we used the AdamW optimizer and cosine annealing rate schedule. An initial learning rate of $2 \times 10^{-4}$ was applied in the Potsdam dataset, and $1 \times 10^{-4}$ was applied in the GID-15 dataset. A momentum decay of 0.0001 was applied in both of the above datasets. The number of training epochs was 300. For data augmentation, 5–15% random linear stretching, fixed angle rotation, and image inversion were used.

### 4.2. Accuracy Assessment

For the experiments evaluated on test datasets, we reported the number of parameters, floating-point operations per second (FLOPs), and frames per second (FPS) of the model. In addition, the intersection over union (IoU) was reported as metrics on the GID-15 dataset, and F1-score and overall accuracy (OA) reported on the Potsdam dataset. mIoU is a semantic segmentation metric that calculates the average of the intersection and concatenation ratio for all categories. The F1-score is the harmonic mean of the precision (P) and recall (R). The core idea of the F1-score is to improve P and R as much as possible so that the difference between them is as small as possible. A higher F1-score value indicates a better segmentation performance of the network. OA indicates the global accuracy, regardless of category, considering only how well all samples are classified, i.e., all correctly classified samples divided by the total number of samples. As shown in Table 1, we reported the main metrics used to evaluate the model, and on both the Potsdam and GID-15 datasets our model shows higher scores than the previous state-of-the-art model PFNet in all metrics.

**Table 1.** Comparison of the main metrics with PFNet on the test set.

| Dataset | Method | mIoU (%) | mF1 (%) | OA (%) | Params |
|---------|--------|----------|---------|--------|--------|
| Potsdam | PFNet | 85.8 | 92.3 | 92.3 | 10.5 M |
|         | ours  | 86.2 | 92.6 | 92.4 | 8.5 M  |
| GID-15  | PFNet | 60.2 | 71.6 | 82.2 | 33 M   |
|         | ours  | 62.5 | 77.1 | 83.2 | 8.5 M  |

### 4.3. Ablation Studies

**Baseline.** To evaluate the performance of the proposed method, we adopted our pixel decoder module SFP as the segmentation head and upsampled its last stage to obtain the results. Therefore, we compared the pixel decoder module SFP with the FPN model. As shown in Table 2, although the result of the mean intersection (mIoU) from the SFP model is only 84.3%, 1.1% less than that from the FPN model; however, when we add cross-attention to them, the performance of SFP is better than FPN. This indicates that the spatial feature information gained from the concatenate design at the high-resolution stages of SFP has improved the network structure's performance.

**Table 2.** Comparison with the FPN pixel decoder.

| Settings | mIoU (%) | Params | GFLOPs |
|----------|----------|--------|--------|
| Baseline | 84.3 | 4.8 M | 18.9 |
| FPN | 85.4 | 6.6 M | 43.7 |
| FPN + cross-attention | 85.2 | 10.6 M | 73.8 |
| ours | 86.2 | 8.5 M | 44.1 |

GFLOPs: Giga FLOPs.

**Learnable Object Embedding Queries (OQE).** We defined a learnable object embedding query as a feature embedding a class of objects and performed an object-oriented feature augmentation on the final output by this query. The number of queries has a large impact on the result of the segmentation. Figure 7 shows our model trained with different numbers of queries on the Potsdam and GID-15 datasets. We find that our model performed best on both datasets when the number of queries was set to approximately 20. This indicates that we do not need to adjust the number of queries in different types of remote sensing datasets. Moreover, the object embedding queries can also enhance performance even without the transformer module. As shown in Table 3, even if we replaced the transformer encoder with a dot product operation and removed the transformer decoder, the performance of the model is still better than the baseline. The mIoU improved by 1.1 when the transformer decoder is added. The mIoU increased by 1.9 when the transformer encoder was also added.

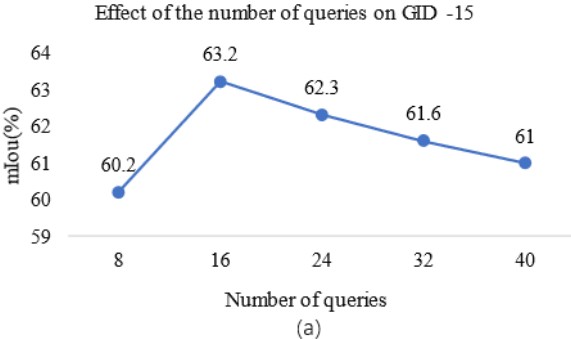
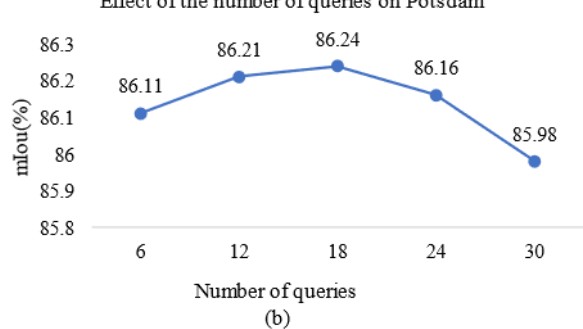

**Figure 7.** Ablation studies on the number of object embedding queries.

**Table 3.** Effect of each component in our model.

| Object Embedding Queries | Transformer Decoder | Transformer Encoder | mIoU (%) | GFLOPs |
|:---:|:---:|:---:|:---:|:---:|
| - | - | - | 84.3 | 18.9 |
| ✓ | - | ✗ | 85.1 | 22.0 |
| ✓ | - | ✓ | 85.3 | 27.4 |
| ✓ | ✓ | - | 85.4 | 32.1 |
| ✓ | ✓ | ✓ | 86.2 | 44.1 |

✗ means that the dot product replaces the transformer encoder.

**Transformer Decoder (TrDec.).** As the module that contributes the most to the model, we tested their significance by removing components from the transformer decoder one at a time. As shown in Table 4, the model performance without cross-attention is lower than that of the baseline. Table 5 shows that our model benefited from multiscale feature resolution and mask attention. Compared with single resolution features (e.g., a single scale of 1/8), the addition of multiresolution features can strengthen the model's performance, and the attention mask kept the mIoU from increasing the inference time. Additionally, self-attention and FFN in the transformer decoder are also effective in enhancing the representation of object embedding queries after cross-attention. The introduction of different resolution features will effectively improve the model performance.

**Table 4.** Effect of the component in the transformer decoder.

| Settings | mIoU (%) |
|:---:|:---:|
| ours | 86.2 |
| self attention | 85.7 |
| cross-attention | 83.9 |
| attention mask | 85.7 |
| all 3 components | 85.3 |

**Table 5.** Effect of different resolution features.

| Settings (in Transformer Encoder) | mIoU (%) | GFLOPs |
|:---:|:---:|:---:|
| single scale of 1/32 | 84.7 | 40.1 |
| single scale of 1/16 | 85.2 | 42.8 |
| single scale of 1/8 | 86.0 | 45.7 |
| multiscale with attention mask(ours) | 86.2 | 43.1 |

**Transformer Encoder (TrEnc.).** In our proposed model, the transformer encoder was used to enhance the representation of semantic features. We used a similar approach as the one for the transformer decoder module for the transformer encoder component validation. As shown in Table 6, the encoder cross-attention (including the subsequent FFN) can aggregate object region representations more effectively than the dot product. Figure 8 shows the gradient changes of the features of different objects that have been aggregated by the transformer module.

**Table 6.** Effect of the component in the transformer encoder.

| Settings | mIoU (%) |
|:---:|:---:|
| **ours** | 86.2 |
| **cross-attention** | 84.6 |
| **ffn** | 86.0 |
| **all 2 components** | 85.8 |

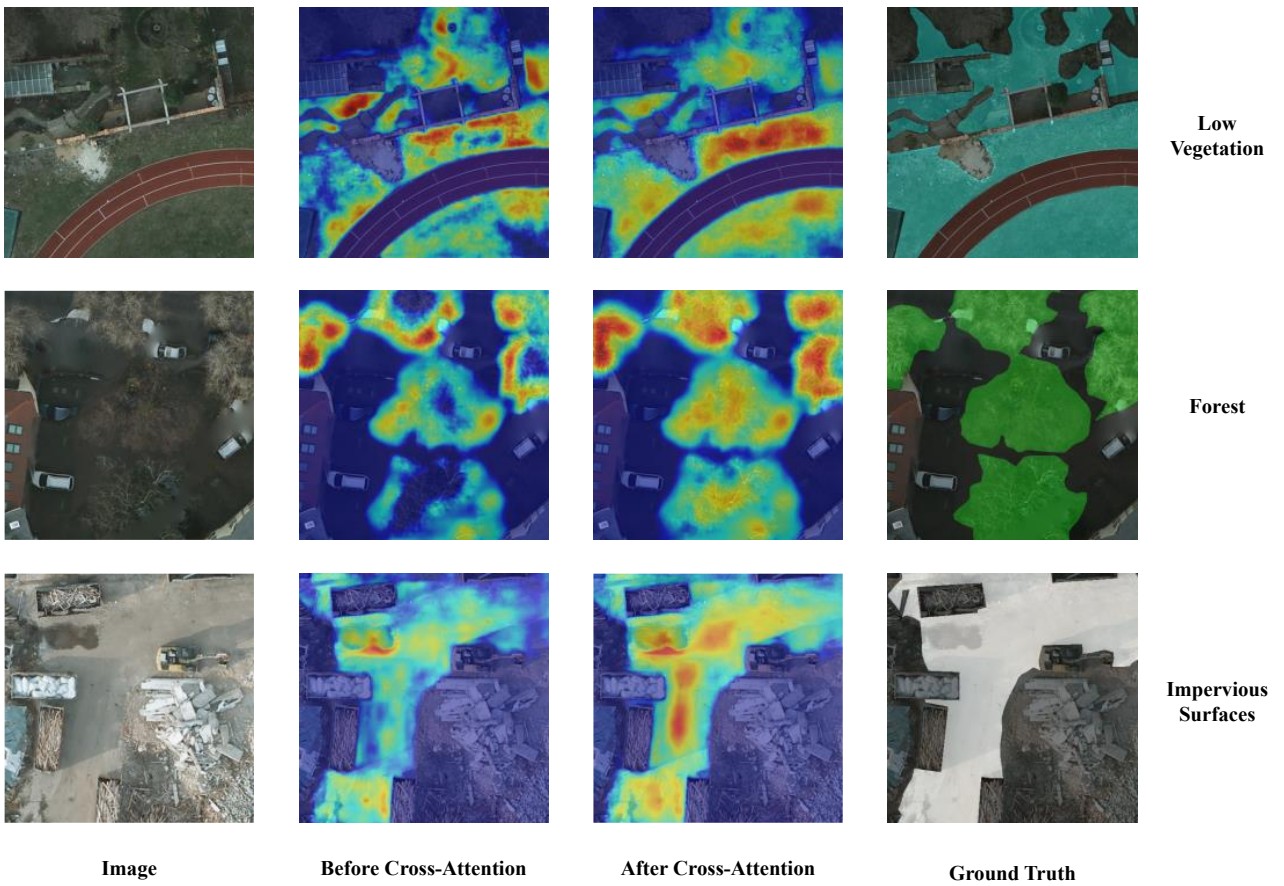

| | | | |
|---|---|---|---|
| | | | Low Vegetation |
| | | | Forest |
| | | | Impervious Surfaces |
| Image | Before Cross-Attention | After Cross-Attention | Ground Truth |

**Figure 8.** Examples of class activation maps (CAM) on the Potsdam validation set.

**Auxiliary Segmentation Head.** We added the transformer encoder as an auxiliary segmentation head to the SFP at different stages. Table 7 illustrates the analysis output for experiments in which the insertion of the auxiliary segmentation head in the low-resolution stage helps to improve the model performance. Notably, adding an auxiliary segmentation head to the SFP stage 4 did not seem to get any improvement; therefore, we decided to remove this part to reduce the training time.

**Table 7.** Effect of the auxiliary segmentation head at different stages.

| Stage 1 | Stage 2 | Stage 3 | Stage 4 | mIoU (%) |
|---------|---------|---------|---------|----------|
| - | - | ✓ | - | 85.3 |
| - | - | ✓ | ✓ | 85.3 |
| - | ✓ | ✓ | - | 85.9 |
| ✓ | ✓ | ✓ | ✓ | 86.2 |
| ✓ | ✓ | ✓ | - | 86.2 |

*4.4. Benchmarking Recent Works on Potsdam Dataset*

We also compared it with some semantic segmentation methods [24,47] based on output for the Potsdam test dataset in Table 8. For an objective comparison, we reimplemented some methods using VAN-tiny as the backbone for feature extraction. All methods used patches cropped to 512*512 for inference at a single scale. Our proposed method achieved the best results among all free available models.

**Table 8.** Quantitative comparison of experiment results on the Potsdam test set with the networks whose backbones are Van-tiny or similar alternatives.

| Method | Backbone | Imp. Surf. | Building | Low Veg. | Tree | Car | Clutter | mIoU (B/F) | mF1 (B/F) | OA | Params | FPS |
|---|---|---|---|---|---|---|---|---|---|---|---|---|
| DABNet [48] | - | 89.9 | 93.2 | 83.6 | 82.3 | 92.6 | - | -/79.6 | -/88.3 | 86.7 | - | - |
| DANet [38] | Van-tiny | 91.5 | 93.1 | 86.1 | 85.5 | 91.6 | 66.9 | 79.1/82.5 | 85.8/89.6 | 89.5 | **4.2 M** | 131.6 |
| BiSeNetV2 [37] | - | 91.3 | 94.3 | 85.0 | 85.2 | 94.1 | - | -/82.3 | -/90.0 | 88.2 | - | - |
| PSPNet [14] | Van-tiny | 92.7 | 96.8 | 88.4 | 87.9 | 85.6 | 81.9 | 80.3/82.5 | 88.9/90.3 | 91.2 | 5.3 M | 161.3 |
| SwiftNet [49] | ResNet50 | 91.8 | 95.9 | 85.7 | 86.8 | 94.5 | - | -/83.8 | -/91.0 | 89.3 | - | - |
| FCN [11] | Van-tiny | 92.6 | 96.1 | 88.1 | 87.2 | 90.7 | 70.3 | 80.8/84.3 | 87.6/91.0 | 91.0 | 6.4 M | **172.4** |
| FPN [33] | Van-tiny | 92.9 | 94.6 | 88.0 | 86.7 | 93.2 | 82.7 | 82.5/85.2 | 89.6/91.1 | 91.4 | 6.6 M | 154.3 |
| UperNet [50] | Van-tiny | **93.6** | 97.2 | 89.1 | 88.0 | 92.5 | 83.2 | 83.3/85.7 | 90.6/92.1 | 91.9 | 7.0 M | 149.3 |
| PFNet [24] | Van-tiny | 93.5 | 97.2 | 89.4 | 88.3 | **92.8** | 83.2 | 83.6/85.8 | 90.8/92.3 | 92.3 | 10.5 M | 110.4 |
| ours | Van-tiny | 93.1 | **97.3** | **90.6** | **89.4** | 92.6 | **84.8** | **83.9/86.2** | **91.3/92.6** | **92.4** | 8.5 M | 76.9 |

## 4.5. Comparison with State-of-the-Art Networks

In addition to the comparison of the benchmark work, we also conducted comparative experiments on the Potsdam and GID-15 datasets with previous state-of-the-art semantic segmentation works as shown in Tables 9 and 10. Our method outperformed previous models. It is worth noting that on the Potsdam dataset, even though our results are not superior for the categories of buildings and cars, our method slightly outperformed other methods by at least av1% point in terms of the average F1 in both the hard-to-classify categories (e.g., trees and low vegetation) and in the background, as shown in Figure 9. Similar to the results obtained from the GID-15 datasets, land cover categories that are often overwhelmed by background and other categories (e.g., ponds, garden plots) are identified with much higher accuracy than other models in the experiment and the models in the Learning to Understand Aerial Images (LUAI) 2021 challenge held on ICCV'2021, as illustrated by the results in Figure 10. This illustrates that our model effectively has improved the semantic representation of indistinguishable classes.

**Table 9.** Comparison with the state-of-the-art networks on the Potsdam dataset.

| Method | Backbone | Imp. Surf. | Building | Low Veg. | Tree | Car | Clutter | mIoU (B/F) | mF1 (B/F) | OA | Params | FPS |
|---|---|---|---|---|---|---|---|---|---|---|---|---|
| SETR [22] | VIT-S | 92.9 | 97.0 | 87.8 | 85.9 | 90.6 | 82.1 | 81.2/83.5 | 89.4/90.9 | 91.0 | 73.2 M | 35.7 |
| HRNetV2 [35] | HRV2-W48 | 93.2 | 97.0 | 88.5 | 87.4 | 92.5 | 80.0 | 81.9/84.9 | 89.8/91.7 | 91.4 | 65.9 M | 38.5 |
| DeeplabV3+ [16] | Xception | **93.7** | **97.3** | 89.5 | 88.5 | 92.6 | 83.8 | 83.6/85.9 | 90.9/92.3 | 92.2 | 54.7 M | 31.3 |
| CCNet [41] | ResNet101 | 93.6 | 96.8 | 86.9 | 88.6 | 92.2 | 83.1 | -/84.9 | -/92.4 | 91.5 | - | - |
| UperNet [50] | Swin-S [21] | 93.6 | 97.2 | 89.1 | 88.0 | 92.5 | 83.3 | 83.5/85.9 | 90.6/92.1 | 91.9 | 81.4 M | 7.6 |
| SegFormer [23] | MIT-B2 | 93.6 | 97.3 | 89.2 | 88.2 | 92.5 | 82.8 | 83.2/85.7 | 90.6/92.2 | 92.0 | 27.5 M | 56.9 |
| ours | Van-tiny | 93.1 | 97.3 | **90.6** | **89.4** | **92.6** | **84.8** | **83.9/86.2** | **91.3/92.6** | **92.4** | **8.5 M** | **76.9** |

-: The results were not reported in the original paper. B/F: Background metrics were calculated/Foreground metrics only.

**Table 10.** Comparison with the state-of-the-art networks on the GID-15 dataset. The abbreviations for classes are IDL-industrial land, UR-urban residential, RR-rural residential, TL-traffic land, PF-paddy field, IGL-irrigated land, DC-dry cropland, GP-garden plot, AW-arbor woodland, SL-shrub land, NG-natural grassland, AG-artificial grassland, and Bg-background.

| Method | IDL | UR | RR | TL | PF | IGL | DC | GP | AW | SL | NG | AG | River | Lake | Pond | Bg | mIou (B/F) | Params |
|---|---|---|---|---|---|---|---|---|---|---|---|---|---|---|---|---|---|---|
| Deeplabv3+ [16] | 64.1 | 73.9 | 62.5 | 59.3 | 56.6 | 70.2 | 41.1 | 21.2 | 89.2 | 15.6 | 64.5 | 37.9 | 60.9 | 73.5 | 23.6 | - | -/54.3 | 54.7 M |
| HPSNet [31] | 63.3 | 74.0 | 64.4 | 59.0 | 53.2 | 75.6 | 47.3 | 22.1 | 91.1 | 17.0 | 69.5 | **45.2** | 55.6 | 72.4 | 23.4 | - | -/55.5 | - |
| HRNetV2 [35] | 67.6 | 76.75 | 50.8 | 69.5 | 53.4 | 89.4 | 59.6 | 13.2 | 55.0 | 29.2 | 78.7 | 6.8 | 59.0 | 84.2 | 24.4 | 65.4 | 55.2/54.5 | 65.9 M |
| SERT [22] | 68.7 | 79.3 | 60.7 | 69.6 | 55.7 | **91.5** | 55.1 | 38.8 | 59.5 | 35.3 | 79.5 | 24.5 | 56.6 | 73.6 | 14.1 | 66.0 | 58.0/57.5 | 73.2 M |
| GFI [51] | 72.0 | 77.9 | 71.2 | 65.8 | 57.3 | 83.0 | 61.4 | 29.7 | 70.6 | 27.8 | 70.6 | 21.9 | 62.2 | 78.5 | 24.2 | - | -/58.2 | - |
| LL [51] | 68.8 | 77.3 | 70.8 | 60.9 | 64.6 | 83.0 | 66.0 | 18.1 | **71.3** | 35.3 | 64.3 | 17.7 | 71.7 | **85.4** | 29.8 | - | -/59.0 | - |
| SegFormer [23] | 69.2 | 79.7 | 69.2 | 71.4 | 57.8 | 90.5 | 59.2 | 40.2 | 59.0 | 37.2 | 82.0 | 26.4 | 60.6 | 82.5 | 13.4 | 67.8 | 60.2/59.7 | 27.5 M |
| UperNet [50] | 69.9 | 79.6 | 63.2 | 70.9 | 53.2 | 89.3 | 60.3 | 48.5 | 60.5 | 35.0 | 81.7 | 22.8 | 62.0 | 75.5 | 27.4 | 65.3 | 60.3/60.0 | 81.4 M |
| PFNet [24] | 69.6 | 79.3 | 65.3 | 71.2 | 55.8 | 89.4 | 62.2 | 46.8 | 62.5 | 30.4 | 82.1 | 22.5 | 61.1 | 82.2 | 22.3 | 67.6 | 60.3/60.2 | 33.0 M |
| AAE [51] | **75.6** | **81.1** | **75.5** | **76.6** | 58.6 | 86.7 | **68.6** | 45.2 | 68.5 | 34.0 | **84.7** | 25.3 | **74.1** | 85.2 | 28.5 | - | -/**64.5** | - |
| ours | 71.0 | 77.0 | 69.2 | 73.2 | **59.1** | 89.0 | 59.4 | **54.9** | 59.1 | **39.5** | 78.5 | 27.2 | 64.1 | 81.6 | **35.1** | **69.2** | **62.9**/62.5 | **8.5 M** |

-: The results were not reported in the original paper. B/F: Background metrics were calculated/Foreground metrics only.

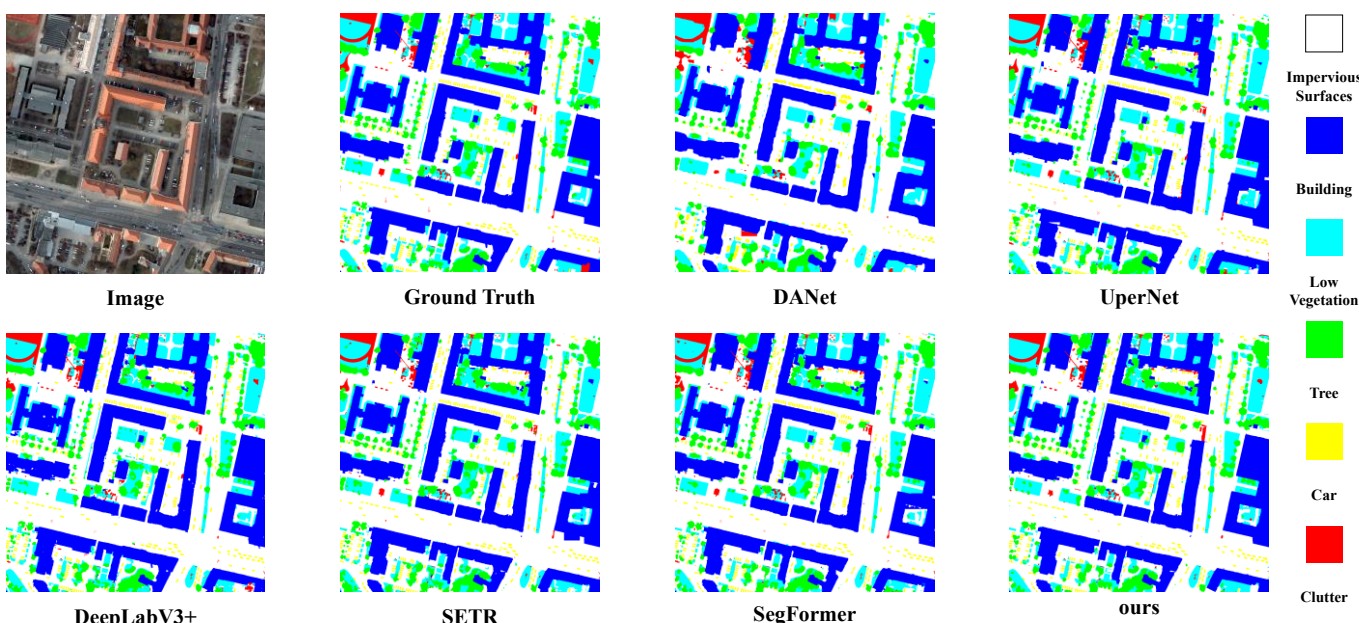

**Figure 9.** Visualization of results from the Potsdam test set.

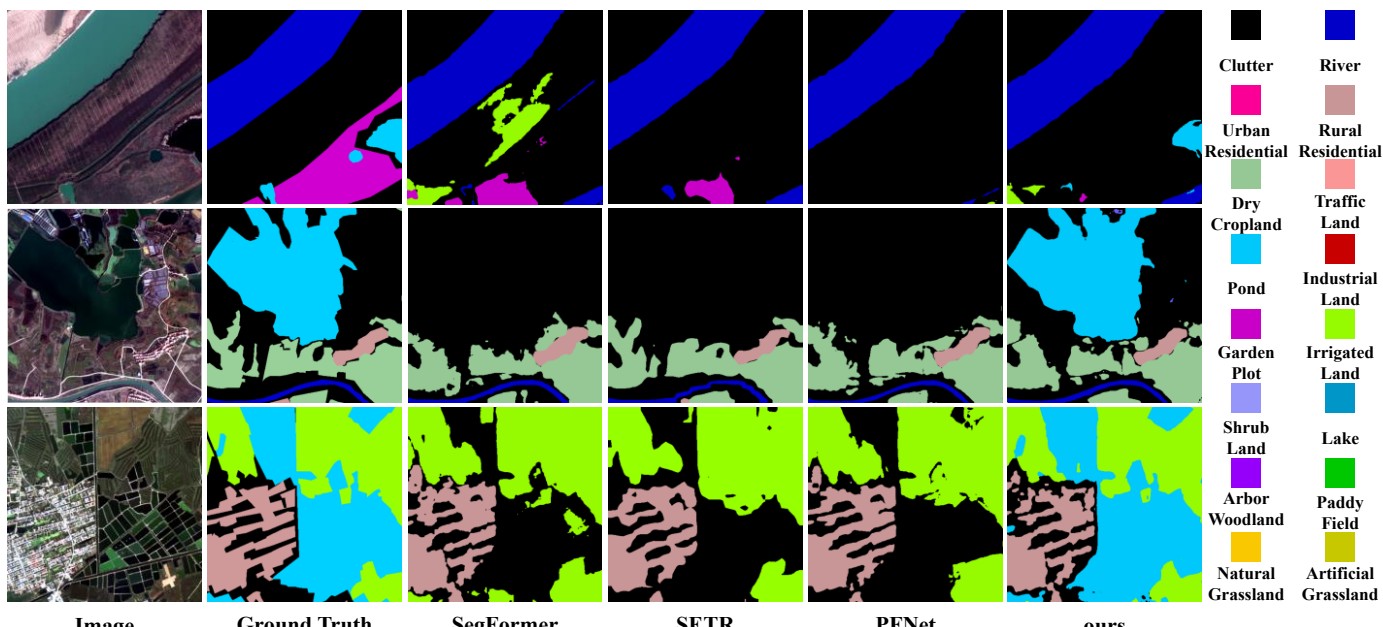

**Figure 10.** Visualization of results for the GID-15 validation set.

### 4.6. The Complexity of the Network

In addition to the segmentation accuracy, the model was also evaluated in terms of the number of parameters and speed. We calculated the FLOPs in the ablation study. The number of computations occupied by each module in the network was evaluated. Furthermore, we reported the number of parameters and frames per second (FPS). As shown in Tables 8 and 9, our model achieved better accuracy than large models, while the inference speed was still similar. Note that the fps was tested for the entire test set on an RTX3080 GPU with a batch size of 1.

## 5. Discussion

Experimental results show that our proposed method can efficiently improve the segmentation accuracy of background and indistinguishable classes. The main contribution

to the accuracy improvement comes from the augmentation of pixel representation by object embedding queries. In Figure 9, notably, the predictions from CNNs [10] backbone [16,24,38,50] show unsmooth boundaries and some salt-and-pepper noise, owing to the loss of object contextual information caused by multiple downsampling. We addressed this problem by hierarchical feature fusion. Compared with FPN [33], UperNet [50], and PFNet [24], we focus more on the high resolution pixel representation, which helps indistinguishable classes to get more object details. Additionally, self-attention [22,38] and dense affinity learning [41] over-introduce features of clutter, and models [35] that rely on these mechanisms to improve model performance have lower accuracy in clutter on both the Potsdam and GID-15 datasets. We addressed this problem by pixel representation augmentation, and achieved the best accuracy in clutter for both datasets. In Figure 10, the three previous state-of-the-art models [23] were unable to identify a pond in the clutter, while our approach successfully identifies them.

Of further importance is how the object embedding queries are generated. As shown in Table 3, the ablation experiments for object embedding queries improve mIoU by 1.1% by using only the transformer decoder without the encoder. So optimizing the generation of coarse segmentation graphs by ground truth [35,39] is not better than random initialization of learnable object embedding queries. A better introduction of feature dependencies (e.g., cross-attention) would help the model performance more.

Furthermore, there is a possible misconception that the model has a large number of parameters and is hard to converge employing the concatenation and transformer modules. In fact, our method achieves convergence at about 120 epochs without cosine annealing, much faster than the pure transformer model [23,25]. Benefiting from efficient encoding of pixel representations by object embedding queries, high-resolution features do not require too many channels of semantics. Under the same backbone, the number of parameters of our method is 8.5 M, which is a little higher than 6.6 M for FPN [33], 7.0 M for UperNet [50], and lower than the 10.5 M for PFNet [24]. With about the same number of parameters, our method achieves the highest mIoU and OA.

## 6. Conclusions

In this study, we have proposed a method based on object embedding queries for land cover classification of multi-resolution remote sensing images. Our main goal was to augment the semantic segmentation network to recognize feature types in complex backgrounds and hard-to-classify categories. To achieve this goal, we have used the following settings: (1) We set up a set of object feature embedding queries within the network and augment the feature representation of each pixel. (2) We redesigned the cross-attention module in the transformer for encoding and decoding different classes of image representation. (3) We trained object embedding queries in multilevel features and output auxiliary losses to improve the learning of spatial features. Our experiments show that the above approach effectively extracts category object representations from multiscale pixel representations thereby helping to identify similar categories and backgrounds. Additionally, we designed the SFP based on FPN with a compressed pixel decoding module to minimize the number of parameters and inference time of the model while ensuring there was no loss of performance.

Our work on the Potsdam and GID-15 test datasets showed strong performance, proving that the cross-attention-based transformer can be effectively applied to remote sensing imagery tasks. However, there are still limitations to our method. An SPF stage5 high-resolution feature map as a query in a transformer encoder takes more calculation resources to compute cross-attention and high-dimensional feature mapping; and the more calculation resources it takes, the higher the number of channels from object embedding queries. This is why we set the number of channels in the feature pyramid to 256 and do not use a large backbone. We will investigate how to augment pixel representation with a high number of channels more efficiently in future work.

**Author Contributions:** Conceptualization, Y.L. and J.W.; methodology, Y.L.; data curation, Y.L. and Z.T.; formal analysis, X.Y. and J.W.; writing original draft preparation, Y.L. and Z.Y.; writing review and editing, Y.L. and J.W.; visualization, Y.L.; supervision, J.W. and X.Y; project administration, J.W.; funding acquisition, J.W. All authors have read and agreed to the published version of the manuscript.

**Funding:** This research was funded by the National Key R&D Program of China, grant number 2021YFE0117300.

**Data Availability Statement:** Not applicable.

**Acknowledgments:** The authors express profound gratitude to ISPRS for availing of the Potsdam dataset and Wuhan University for availing of the GID-15 datasets used in our research.

**Conflicts of Interest:** The authors declare no conflict of interest.

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
