# Peer review of "Pixel Representation Augmented through Cross-Attention for High-Resolution Remote Sensing Imagery Segmentation"

_remotesensing, doi:10.3390/rs14215415_

Round 1

Reviewer 1 Report

The paper presents an exciting approach to pixel augmentation for semantic segmentation. However, the novelty and significance of the performance compared to others are not well established. Mainly the gain in performance for different classes and number of parameters is not put performing others in most cases. 

Moreover, Figures 5 and 10 are extremely hard to follow. Please look into them to improve the readability and explainability. In the discussion part, the limitations of the methods could be more elaborately explained.

Reviewer 2 Report

This paper proposes a CNN-based network for pixel-wise classification of remote sensing images. To overcome the issues of complex background and interclass similarity occurred in remote sensing images, the authors augment the semantic segmentation network by introducing many modern CNN techniques such as feature embedding queries and cross-attention module. The experiments conducted on two real remote sensing images show the effectiveness and efficiency of the proposed methods.

The motivation, background, and related works were well introduced. The ablation study is also included. Therefore, the contribution is apparent. However, the reviewer thinks there are few issues that need to be addressed to meet the publication criterion. The comments are as follows:

The comments:

1. In order to clearly see the advantages of the proposed method, it is recommended to use bold fonts indicate the highest (or lowest, i.e., FPS) values in each column in Table 2, Table 3, and Table 4.

2. The authors use an image scene in Figure 1 to illustrate “complex background” and “interclass similarity”. How is the definition of complex background and interclass similarity? I think the authors need to do a more detailed explanation in the figure caption.

3. According to abstract, the proposed method can deal with the issues of complex background and interclass similarity. But the experimental section seems to lack a discussion of these two points. The authors have to compare the segmentation map results of state-of-arts methods, and select some local regions to validate that the proposed method can overcome these two issues.

4. In Section 4.5 (Comparison with state-of-arts networks), the authors should list the names of state-of-art methods with their reference numbers. Also, in the quantification results shown in Table 5, Table 6, and Table 7, it is also recommended to mark each state-of-art method with a ref. number.

5. Figure 2 demonstrates the architecture of the proposed method. Obviously, the input is missing, and there is no Stage 4 in SFP module.

6. In the beginning of Section 3, the authors define the problems of land cover classification and semantic segmentation. The font of the used notations seem not be well unified. For example, the authors used two fonts for y and p. In lines 227 and 228, H, W, and K_{i} are bold and italic. They are not matrices or vectors so should not be bold. Beside, y and y^{gt} are not defined. Please define those notations in the revised version.

7. The table order is wrong. There are two Table 2, Table 3, and Table 4.

Reviewer 3 Report

Dear Authors,

thank you very much for your interesting manuscript, I like your idea, the first part of the manuscript (introduction and the state-of-the-art) is significantly too general, you need to add much more details, including description of used methods. But the most weak element of the manuscript is your Discussion. It is very important part how authors compare their own achievements with references, what is better/worse and why? You need to develop the part much deeper adding around 40% of all references to the chapter.

Please, look at my comments, which are attached to the manuscript.

best regards

Reviewer

Round 2

Reviewer 2 Report

The authers have well handles the reviewer's comments. 

Author Response

Thank you for recognizing us!

Reviewer 3 Report

Dear Authors,

your revision is not enough deep, in many parts you added few sentences, which do not improve the quality of the manuscript.

The most important elements, which need a deeper revision are:

- too general state-of-the-art, there are too many well-known statements,

- accuracy assessment and validation of used models,

- Discussion, you need to compare your more important results whith references to highlight your more innovative solutions.

Much more detailed comments are presented in the attached manuscript.

best regards

Reviewer

Round 3

Reviewer 3 Report

Dear Authors,

the manuscript looks much better, but please, add much more references to the Discussion to compare your more important achievements with references highlighting your novelity. The Discussion is still too general.

Best regards

Reviewer

Author Response

Dear reviewer,

Thank you very much for your comments on the revision of our paper, we have modified the Discussion based on our experimental results and previous methods to provide a strong basis for our novelty. Differently from the last version, we have reduced the description of the principles and focused on the description of the experimental results and the advantages over the previous methods.
